# The Biological Activity of Fragmented Computer-Aided Design/Manufacturing Dental Materials before and after Exposure to Acidic Environment

**DOI:** 10.3390/medicina59010104

**Published:** 2023-01-03

**Authors:** Codruța Eliza Ille, Elena-Alina Moacă, Maria Suciu, Lucian Barbu-Tudoran, Meda-Lavinia Negruțiu, Anca Jivănescu

**Affiliations:** 1Department of Prosthodontics, Faculty of Dental Medicine, “Victor Babes” University of Medicine and Pharmacy, Revolutiei Ave. 1989, No. 9, 300041 Timișoara, Romania; 2TADERP Research Center—Advanced and Digital Techniques for Endodontic, Restorative and Prosthetic Treatment, “Victor Babeș” University of Medicine and Pharmacy, Revolutiei Ave. 1989, No. 9, 300041 Timişoara, Romania; 3Department of Toxicology, Drug Industry, Management and Legislation, Faculty of Pharmacy, “Victor Babeș” University of Medicine and Pharmacy Timisoara, 2nd Eftimie Murgu Square, 300041 Timişoara, Romania; 4Research Centre for Pharmaco-Toxicological Evaluation, “Victor Babeș” University of Medicine and Pharmacy, 2nd Eftimie Murgu Square, 300041 Timișoara, Romania; 5Electron Microscopy Laboratory “Prof. C. Craciun”, Faculty of Biology & Geology, “Babes-Bolyai” University, 5-7 Clinicilor Street, 400006 Cluj-Napoca, Romania; 6Electron Microscopy Integrated Laboratory, National Institute for R & D of Isotopic and Molecular Technologies, 67-103 Donat Street, 400293 Cluj-Napoca, Romania; 7Department of Prostheses Technology and Dental Materials, Faculty of Dental Medicine, “Victor Babes” University of Medicine and Pharmacy Timişoara, Revolutiei Ave. 1989, No. 9, 300041 Timişoara, Romania; 8Research Center in Dental Medicine Using Conventional and Alternative Technologies, Faculty of Dental Medicine, “Victor Babeş” University of Medicine and Pharmacy Timişoara, Revolutiei Ave. 1989, No. 9, 300041 Timişoara, Romania

**Keywords:** occlusal veneers, CAD/CAM restorative materials, artificial acidic saliva exposure, cytotoxicity, human fibroblast, human keratinocytes, cell adherence

## Abstract

Three ceramic and composite computer-aided design/computer-aided manufacturing (CAD/CAM) materials from different manufacturers (Cerasmart (CS)—nanoceramic resin; Straumann Nice (SN)—glass ceramic and Tetric CAD (TC)—composite resin) were tested to investigate the biocompatibility and sustainability on human fibroblasts and keratinocytes cells. Each type of CAD/CAM blocks restorative materials with fine and rough surfaces was exposed to an acidic environment for one month. After that, various powders were obtained by milling. In parallel, powders were also prepared from each restorative material, which were not exposed to the acidic environment. The cytotoxic effects were investigated by means of MTT and LDH assays, as well as nitric oxide production on two human normal cell lines, namely, fibroblasts (BJ) and keratinocytes (HaCaT). In addition, the degree of adhesion of fibroblast cells to each CAD/CAM material was evaluated by scanning electron microscopy (SEM). The results showed that the two samples that were exposed to an acidic environment (CS and SN) induced a reduction of mitochondrial activity and plasma membrane damage as regards the fibroblast cells. A similar effect was observed in TC_fine-exposed material, which seemed to induce necrosis at the tested concentration of 1 mg/mL. No oxidative stress was observed in fibroblasts and keratinocytes treated with the CAD/CAM materials. Regarding the adhesion degree, it was found that the fibroblasts adhere to all the occlusal veneers tested, with the mention that the CS and SN materials have a weaker adhesion with fewer cytoplasmic extensions than TC material. With all of this considered, the CAD/CAM restorative materials tested are biocompatible and represent support for the attachment and dispersion of cells.

## 1. Introduction

As novel dental materials appear on the market, clinical reactions such as pulpal or gingival irritation remain a cause for concern. Even though conventional tests are employed in the early phases of a material’s development, a postmarket information system that allows dentists to report adverse effects should be added to the biocompatibility assessment [1]. The introduction of novel materials and techniques [2,3,4], as well as the advancement of CAD/CAM (computer-aided design/computer-assisted manufacturing) technology, have significantly altered the clinical workflow in the field of dentistry, leading to new concepts for evaluation and dental treatment [5]. The CAD/CAM technique has the great advantage of reducing production time and achieving transposed and perfectly adapted structures [6,7,8,9,10,11]. In a research study, three groups of materials were explored as regards their physical properties, and the authors concluded that 3D-printed interim specimens showed improved fit, adaptation, and wear properties compared to CAD/CAM, as well as conventional interim restorations [12]. 

Among various CAD/CAM materials for dental restorations, the two most intensively studied classes are ceramic and composite resin. CAD/CAM ceramic materials present superior mechanical and aesthetic properties, while CAD/CAM composite resins present specific significant advantages related to their machinability and intraoral reparability [13,14]. Minimally invasive dentistry is a suitable solution that may convince more patients to start treatment when erosion or attrition processes are still in their early stages. According to recent data, lithium disilicate/silicate ceramics and high-performance CAD/CAM composite resins should be used to create ultrathin occlusal devices [15,16]. Regarding their biocompatibility, a study reported that ceramic materials offer better biocompatibility and cell response compared to polymers [17]. Many improved materials with superior mechanical qualities have been developed for CAD/CAM technology in the last decade [18,19,20,21,22,23,24,25]. With the development of novel nanomaterials, the standards of dental care health also improved by altering the properties of biomaterials [26]. Regarding long-term treatment strategies, dental clinicians desire biocompatible dental materials that are aesthetically pleasing and have good mechanical features [27,28]. In order to select the most suitable material, some important attributes must be known, e.g., material composition, surface roughness, mechanical features, ability to retain biofilm, and interaction with the oral environment [29].

Nanodentistry is considered a field with huge potential, especially in restorative dental science, by giving rise to personalized treatment [30,31,32]. Nanoparticles have been employed to improve the mechanical properties of dental composites, enhancing their bonding and anchoring, and lowering friction. Smaller particles penetrate better into deeper lesions and reduce dental composite porosities for greater mechanical strength. Greater bioactivity, including bonding and integration, as well as more powerful action against microbes, are made possible by the increased surface area-to-volume ratio. Encapsulated bioactive compounds, including drugs and growth factors, can be supplied more accurately and with site-targeted delivery for localized treatments through controlled release. The reengineering of dental prostheses and braces, as well as periodontology, endodontics, and other areas of dentistry, have all benefited from these qualities [33].

There are few research studies regarding the biocompatibility of these types of CAD/CAM restorative materials reported in the scientific literature. For instance, a recent study showed data regarding the biocompatibility and sustainability of two resin-based composites, namely Cerasmart (CS) and Brilliant Crios (BC), and one hybrid ceramic, namely Vita Enamic (EN). In terms of surface roughness, biofilm formation, cytotoxicity, genotoxicity, and cellular changes observed under transmission electron microscopy (TEM), the authors concluded that the investigated CAD/CAM blocks showed no significant difference in surface roughness. Moreover, no correlation was shown between surface roughness and biofilm formation. Considering cytotoxicity, BC showed the highest values, followed by CS and EN. Consequently, EN was considered the most biocompatible material among the tested ones [34]. Another study reported the biocompatibility of Tetric CAD and eight other resin materials [35]. The outcomes reported by the authors showed that both resin blocks for subtractive manufacturing presented the best cell viability results among all tested materials [35]. Daguano and coworkers [36] designed and produced a novel lithium silicate glass ceramic and evaluated its biocompatibility in vitro. The most important discovery was that the new lithia-silica glass ceramic is bioactive, in contrast to the parent glass and other glass ceramics that already exist in the same family. It stimulated MG-63 cells to create a bone-like matrix, which may be very important for bone regeneration in orthopedic and dentistry applications. It also encouraged cell adhesion and proliferation [36].

The tooth-restoration hybrid remains exposed to oral conditions for a long time, therefore attention must be paid to the effect of the intrinsic and extrinsic substances on these surfaces [37]. Patients who have gastroesophageal reflux are the most vulnerable in this situation because they frequently have gastric acid from the stomach flush into the esophagus or mouth cavity [38]. Ceramic dental materials are resistant to chemical attack, however, the pH of the medium, the length of exposure, and the exposure temperature may have an impact on their composition [39]. The components of these materials are transported through saliva, drinks, or food and then swallowed to reach to the gastrointestinal tract [40,41]. That is why it is necessary to evaluate the biocompatibility of these materials even after a certain period of time, during which they were exposed to various factors that could have modified their composition. Moreover, the biocompatibility assessment represents a mandatory stage in the evaluation of the cytotoxicity of dental material, due to the fact that these materials are unstable in a moist oral environment [42].

In accordance with all mentioned above, the purpose of the present study was to compare the early response and biocompatibility of three dental materials in contact with human gingival fibroblasts and human keratinocyte cells. Thus, it was desired to investigate the effect an acidic environment could have on the composition and structure of the three types of CAD/CAM restorative materials, more precisely if acidic artificial saliva could influence the biocompatibility of restorative materials in contact with living organisms. As regards the biological activity, the cytotoxicity of restorative materials (Cerasmart—CS; Straumann Nice—SN and Tetric CAD—TC) on two human normal cell lines—fibroblasts cells (BJ) and keratinocytes (HaCaT), was evaluated. The cytotoxicity was assessed using the MTT proliferation test (3-[4,5-dimethylthiazol-2-yl]-2,5 diphenyl tetrazolium bromide), and LDH assay (lactate dehydrogenase). In addition, the nitric oxide (NO) production assay was performed. Further, the degree of adhesion to normal fibroblasts of occlusal veneers not exposed to acidic artificial saliva was evaluated. The null hypothesis refers to the fact that there will be no difference in the cytotoxicity levels of the BJ and HaCaT cells in contact with the CAD/CAM restorative materials exposed and not exposed to an acidic environment. Moreover, the second null hypothesis is that there will be no difference regarding the NO oxidative stress given by the three materials on both human cell lines. In summary, the current research aims to investigate in depth the biocompatibility of three chairside materials qualified to be used for the minimally invasive treatment of dental wear, previously prepared as CAD/CAM occlusal veneers.

## 2. Materials and Methods

### 2.1. Materials Used and Preparation Procedure

The CAD/CAM restorative materials used in the current study were presented in Table 1, together with their material classification, composition, and batch number. The CAD/CAM blocks were acquired from the following manufacturers: Cerasmart (GC Europe Dental Products, Tokyo, Japan); Straumann Nice (Nice, Straumann Freiburg, Germany) and Tetric CAD (Ivoclar Vivadent, Schaan, Liechtenstein).

Due to the fact that most patients suffer from gastroesophageal reflux and may have damaged occlusal veneers, one of the objectives of the current research was the biological evaluation of the above-mentioned CAD/CAM restorative materials. In order to better simulate the restorative material’s life cycle in the oral environment, all the samples were immersed in an artificial acidic environment (acidic artificial saliva) and kept at 37 °C for 1 month.

From each CAD/CAM restorative material block, three types of samples were prepared, as follows:-Samples denoted with A, from each finished CAD/CAM block not exposed to acidic artificial saliva, a powder was obtained using a straight handpiece and a diamond-coated disc double side (Komet USA LLC, Rock Hill, SC, USA). After that, the CAD/CAM block surface remained rough.-Samples denoted with B, the CAD/CAM blocks with rough surfaces, which remained after samples A were obtained, were immersed in acidic artificial saliva and after 1 month were milled with the diamond-coated disc double side until a powder was obtained;-Samples denoted with C, other finished CAD/CAM blocks from each group, were immersed in acidic artificial saliva, kept for 1 month, and milled with the diamond-coated disc double side, until fine powders were obtained.

Images with CAD/CAM blocks of finished and rough surfaces are depicted in Figure 1, together with the powders and slices obtained from them. The fine surface of the CAD/CAM restorative materials was obtained by using a finishing and polishing kit (Ceramic Finishing Kit—LD2748) from Komet (Rock Hill, SC, USA).

Thus, we wanted to investigate the in vitro potential cytotoxic effect of this CAD/CAM restorative materials (powders and slices), exposed and nonexposed to acidic artificial saliva, on BJ and HaCaT cells (Table 2).

### 2.2. Artificial Acidic Saliva Preparation and Samples Exposure

The protocol used for the preparation of acidic artificial saliva was extensively described in the previous study published by our research group [43]. Briefly, the artificial saliva was obtained by weighing NaCl, KCl, CaCl_2_·2H_2_O, and CO(NH_2_)_2_ in a mass ratio of 1:1:2:2.5 and dissolved in 1 L of distilled water, until a clear solution was obtained. By using the Thermo Scientific Eutech pH 150 portable pH meter, with an electrode (Thermo Scientific, Waltham, MA, USA), the pH solution was measured, and the value obtained was 7.059 at 23° ± 1 °C. The acidic artificial saliva was obtained by adding HCl 37% until the pH was lowered to a 2.99 value at 23° ± 1 °C. The acidic artificial saliva was kept in the refrigerator until future use.

The samples with fine and rough surfaces were immersed in the as prepared artificial acidic saliva and kept in an orbital stirrer incubator (ES20/60 Biosan, Riga, Latvia), for 1 month, at 37 °C to better simulate the oral environment [44]. After 1 month, the CAD/CAM blocks were removed from the acidic environment, washed several times with distilled water, and left to dry at room temperature, after which they were subjected to milling until powders were obtained.

### 2.3. Structural and Morphological Investigations

The powders obtained from all three types of CAD/CAM restorative materials (rough exposed, fine exposed, and nonexposed) were subjected to electron microscopy characterization. The morphological and structural changes induced by the acidic artificial saliva (pH = 2.99) on the exposed CAD/CAM materials, as well as the morphology and structure of the nonexposed samples, were assessed by scanning electron microscopy (SEM), by using a Hitachi SU8230 cold field emission gun STEM (Chiyoda, Tokyo, Japan) microscope with EDX detectors X-Max^N^ 80 from Oxford Instruments (Bristol, United Kingdom). The SEM images were recorded at two magnification orders, one for a general overview of the image/measurements (×100) and another for higher surface topography (×5.00 k). The elemental composition of each CAD/CAM restorative material powder (exposed or nonexposed to artificial acidic saliva) was assessed by EDX analysis and the identified chemical species were expressed in weight percent (wt %).

### 2.4. Cell Cultures Media and Cell Lines

Normal human fibroblasts (BJ cell line, acquired from ATCC (American Type Culture Collection, CRL2522)) and normal human keratinocytes (HaCaT cell line, from CLS—Cell Line Services) were grown in DMEM media (high glucose Dulbecco’s Modified Eagle’s Medium) supplemented with 10% fetal calf serum, 1% L-glutamine, and 1% penicillin-streptomycin and kept at 37 °C, 5% CO_2_, and saturated humidity. The powders obtained from CAD/CAM restorative material blocks were kept for 24 h in a cell culture medium at 37 °C, 5% CO_2_, and saturated humidity at 1 mg/mL concentration prior to their testing (ISO 10993-5:2009). The powders were used as such and diluted to 0.5 and 0.2 mg/mL on cell cultures.

### 2.5. MTT Protocol—Mitochondrial Activity Assessment

Cellular viability was quantified via colorimetric assay using MTT cell viability reagent (3-[4,5-dimethylthiazol-2-yl]-2,5 diphenyl tetrazolium bromide). The MTT viability test measures cellular viability through mitochondria metabolism. The MTT colorimetric test was employed as previously described [45]. In brief, cells were seeded in 96-well plates at 10^4^ cells/well and left to attach and expand for 24 h. Cells were exposed to the restorative materials for 24 h and then MTT was added at a 0.5 mg/mL final concentration. After 1.5 h, the formazan was solubilized with acidified propanol, and absorbance was read at a wavelength of 550 nm (and 630 nm for background) on BioTek Epoch Microplate Spectrophotometer (Agilent Technologies, Santa Clara, CA, USA). Cellular viability of 100% was attributed to control wells, where cells were cultured without CAD/CAM restorative material powder.

### 2.6. LDH Release Method—Cytotoxicity Test

The cytotoxic potential of the tested samples was investigated using the LDH release test, as previously described [46]. In brief, 50 µL of each treated cell culture media was added to a separate 96-well plate together with a 50 µL phosphate buffer, 50 µL Li lactate, and 50 µL NAD solution. The mixture was immediately read by the spectrophotometer at a wavelength of 490 nm and 630 nm, and then again after 5 min. The LDH release was calculated in mU/min of enzymatic activity.

### 2.7. NO by Griess Assay

The in brief protocol applied was 50 µL of the cell culture media obtained after treatment was mixed with 50 µL of sulfanilamide solution and kept for 10 min in the dark, then 50 µL of NED was added and kept for another 10 min in the dark. The mixture was read at a wavelength of 560 nm and the concentration was calculated based on a standard curve for µg concentrations of nitrites and nitrates [47].

### 2.8. Adherence Test by Scanning Electron Microscopy

In order to perform the cells adherence test, from each finished CAD/CAM restorative material block (CS, SN, and TC), which were not exposed to an acidic environment, slices were obtained with a thickness of 0.1 mm (Figure 1). Cells were seeded on the dental materials slices and were left to attach and grow for 24 h. The CAD/CAM restorative materials slices were fixed with glutaraldehyde 2.7% for one hour, washed with PBS for 15 min, and then with pure water for 15 min. After that, the CAD/CAM slices were left to dry for 24 h and then were sputter coated with a layer of 10 nm Pt/Pd. Coated CAD/CAM slices were analyzed with a Hitachi SU8230 Scanning Electron Microscope (Chiyoda, Tokyo, Japan) at 30 kV and 10 µA.

### 2.9. Statistical Analysis

For data collection and statistical analysis Excel 2013 software was used. Data are presented as the mean of three independent experiments ± Standard Deviation (SD). Student’s *t*-test was employed to determine the statistical differences.

## 3. Results

### 3.1. The Structural Characterization of the Nonexposed and Exposed CAD/CAM Restorative Materials Blocks to Acidic Artificial Saliva

In Figure 2 are depicted the SEM images of CS restorative material powder: nonexposed (A), rough exposed (B), and fine exposed (C) at different magnification orders (×100 and ×5.00 k). As one can see, a slight difference is observed between CS_B and CS_C powders, exposed to acidic artificial saliva. The rough powder exposed to acidic artificial saliva has a more expanded appearance, being slightly more voluminous.

In Figure 3 are depicted the SEM images of SN restorative material powder: nonexposed (A), rough exposed (B), and fine exposed (C), at different magnification orders (×100 and ×5.00 k).

Straumann Nice restorative material appears more compact than Cerasmart (Figure 2). Even more, including that milling the material to obtain powder was not successful, the powder apparently still looks compact. A slight granulation can be observed in the powder SN_A, which was not exposed to acidic artificial saliva (image at 10 µm scale). It is possible that long-term exposure to acidic artificial saliva leads to the cementation of this type of material, due to the phase composition and the fact that this CAD/CAM restorative material is classified as a glass ceramic one.

In Figure 4, are depicted the SEM images of TC restorative material powder: nonexposed (A), rough exposed (B), and fine exposed (C), at different magnification orders (×100 and ×5.00 k).

Apparently, this CAD/CAM restorative material resembles the Cerasmart material (Figure 2) quite well, perhaps due to the fact that both are classified as resin-based materials. Cerasmart is nanoceramic resin based, and Tetric CAD is composite resin based. The powder TC_B exposed to the acidic artificial saliva appears more voluminous.

Figure 5 shows the chemical composition of CAD/CAM restorative materials powders (CS, SN, and TC), nonexposed and exposed to acidic artificial saliva.

It can be observed that, in all the CAD/CAM powders, regardless of the environment (exposed/nonexposed to acidic artificial saliva), the EDX analysis confirms the presence of oxygen, carbon, and silicon elements. Straumann Nice restorative material contains the most elements, including traces of fluorine, calcium, and barium, in the case of SN_C powder. Traces of fluoride are also contained in the Cerasmart restorative material, regardless of the sample preparation method (CS_A, CS_B, or CS_C). In addition, even sample B of Tetric CAD (rough CAD/CAM block exposed to acidic artificial saliva) contains traces of fluorine in its composition. Making a comparison between the elemental composition obtained from the EDX analysis and the chemical composition given by the manufacturer (Table 1), the appearance of certain extra elements can be observed. Even if they are present in traces, they are considered harmful to the human body. These elements can be considered impurities and can come from the preparation procedure of the CAD/CAM powders, including their milling (the transfer of the elements from the milling diamond-coated disc to the powders, due to the wear of the dental milling disc).

### 3.2. Impact of the CAD/CAM Restorative Materials Powders on Mitochondrial Activity by Means of MTT Assay

In order to evaluate the cytotoxicity of the CAD/CAM restorative materials, cell viability was measured by MTT assay at 24 h evaluation time (Appendix A).

Figure 6 shows the effect exerted by all the CAD/CAM restorative materials on BJ cells. The mitochondrial activity appeared to be significantly influenced when the cells were treated with CS_C powder, milled from the finished CS–CAD/CAM block, after immersion in artificial acidic saliva for one month at 37 °C, as well as with SN_C powder, exposed also for one month to acidic artificial saliva (obtained from the finished SN CAD/CAM block) and all the prepared Tetric CAD restorative materials powders (TC_A, TC_B, and TC_C, which appeared the most significant).

Regarding the CS restorative material, at a concentration of 1 mg/mL, this reduces BJ cell viability to 60–70%, regardless of the form tested, CS_A (nonexposed) and CS-B powders (exposed to acidic artificial saliva). At 0.5 and 0.2 mg/mL tested concentrations, the CS_A and CS_B powders induced a 10–20% reduction in BJ cell viability. The CS_C powder determined the cell viability reduction to below 50% at all the tested concentrations.

Regarding the SN_A and SN_B powders, they obtained a similar result, slightly under 90% viability at all the tested concentrations, with the exception of 0.2 mg/mL concentration, where mitochondrial activity was close to 100%. Again, the SN_C powder exposed to acidic artificial saliva induced a decrease in BJ viability to about 60% at all the tested concentrations. TC_A and TC_B powders presented similar effects, meaning the reduction of BJ cell viability to 60% (at a concentration of 1 mg/mL), ~70% (at a concentration of 0.5 mg/mL) and ~80% at a concentration of 0.2 mg/mL. The TC_C powder, exposed to acidic artificial saliva, had the lowest viability (35–40%) at the highest concentrations (1 mg/mL and 0.5 mg/mL).

In Figure 7 are depicted the effect exerted by all CAD/CAM restorative materials powders on HaCaT cells. One can note that the HaCaT cells were less affected, presenting 80–90% viability (in a concentration-dependent manner) for all the tested CS restorative material powders, having a better response to the bigger concentration tested (1 mg/mL). The SN restorative materials powders had similar responses, with the exception of SN_B at a concentration of 0.2 mg/mL, where viability was reduced to 50%. TC restorative material powders were better accepted only when the TC–CAD/CAM block was exposed to acidic artificial saliva, either with a rough or fine surface, presenting mitochondrial activity close to 100%.

### 3.3. Cytotoxicity Evaluation of CAD/CAM Restorative Materials Powders via the LDH Release Method

In order to complete the preliminary screening tests on BJ and HaCaT cells, the LDH release assay was employed when treating the cells with concentrations of 1 mg/mL, 0.5 mg/mL, and 0.2 mg/mL at 24 h postexposure (Appendix A). The results obtained are presented in Figure 8 and Figure 9. Consequently, LDH release was within the limit of the untreated control (3.8 mU/min) in the case of BJ cells for all the tested CAD/CAM restorative materials powders and concentrations, with the exception of CS_C, SN_C powders (where the LDH was greatly reduced to almost 0), and TC_C powder (at a concentration of 1 mg/mL and 0.2 mg/mL), where LDH was increased by approximately 50% compared to the untreated control.

HaCaT cells released LDH similarly to the untreated control for the case of CS_A, CS_B, and SN_A powders (at concentrations of 1 mg/mL and 0.5 mg/mL). LDH was significantly reduced in the case of CS_C, SN_B, SN_C, and TC_B powders, and even more so for the case of TC_C powder, at all tested concentrations. A slight and significant increase was noted for CS_A, CS_B, and SN_A powders (at a concentration of 0.2 mg/mL), and for all concentrations tested of TC_A restorative material powder.

### 3.4. NO Production via Griess Assay

Nitric oxide measured values were reduced to almost half of the untreated cell culture, for the human BJ cells (0.03 µg/mL) with almost all tested CAD/CAM restorative materials powders and concentrations. The few exceptions, that were slightly higher, also had a larger SD value. For HaCaT cells, the untreated NO release measured 0.015 µg/mL, and all tested CAD/CAM restorative materials powders had similar, or slightly increased values (up to 0.028 µg/mL), again with reduced significance. The results obtained are presented in Figure 10 and Figure 11, as well as in Appendix A.

### 3.5. The Degree of Adhesion of CAD/CAM Restorative Materials Slices, on Human BJ Cells

Due to the fact that BJ cells were more affected (due to the reduction of mitochondrial activity, Figure 6), the adhesion degree to human BJ cells of the nonexposed CAD/CAM blocks to acidic artificial saliva, was evaluated. The morphology and ultrastructure of human BJ cells, after 24 h of adhesion, were observed via SEM. The SEM images, recorded at two magnification orders, one for a general overview of the image/measurements (×200) and another for higher surface topography (×2.5 k), are depicted in Figure 12.

It can be observed that the BJ cells adhere to the CAD/CAM slices, which shows that the restorative materials are biocompatible and represent support for the attachment and dispersion of cells. This aspect indicates a possible positive reaction of the gum that is in contact with the restorative material. However, the BJ cells do not react in the same way to the three tested CAD/CAM restorative materials. CS and SN determined a weaker adhesion with fewer cytoplasmic extensions than TC. With all that, it keeps its fusiform, morphologically normal appearance. At the TC restorative material, it can be seen how the surface of the material is covered in a percentage of 40–50% after only 24 h, the cells being strongly flattened, adhered to the substrate provided by the restorative material, and having numerous extensions typical of fibroblasts, namely filopodia and lamellipodia.

## 4. Discussion

The ability of a material to not affect the local or systemic behavior of a living organism refers to its biocompatibility with that organism. The most important way to verify biocompatibility is to determine the cytotoxicity through in vivo or in vitro studies [48,49].

The present study investigated the human fibroblasts and keratinocytes response by the MTT and LDH cytotoxic methods, as well as the NO oxidative stress on CAD/CAM restorative materials powders, which are widely used as dental veneers.

The powders CS_C, SN_C, and TC_C (obtained by milling the fine surface of their CAD/CAM blocks), which were previously immersed for one month in acidic artificial saliva, induced a reduction of mitochondrial activity in BJ cells. Also, LDH was reduced only for the CS_C and SN_C restorative materials powders. This may be due to increased endocytosis of the material which may have led to cell stasis, apoptosis, or necrosis of the fibroblasts early after being placed in contact with the cells. TC_C restorative material powder seems to induce necrosis at 1 mg/mL, however, not NO oxidative stress, much later than the first two. Following the study, no differences between the samples not exposed (A) to acidic artificial saliva and rough exposed (B) to acidic artificial saliva, in the case of BJ cells.

Keratinocytes mitochondria were not affected by the CAD/CAM restorative materials powders treatment, and this fact correlated with the reduced LDH release for the powders CS_C, SN_C, TC_C, and SN_A, which may indicate two possibilities: the same number of cells with reduced exocytosis and increased endocytosis (due to too many particles attached to the cell surface), or a reduced number of cells with increased mitochondrial activity (due to particle overload). Again, there was no oxidative stress by NO pathways for HaCaT cells treated with the CAD/CAM restorative materials powders.

Therefore, considering the outcomes mentioned above, one can affirm that the first hypothesis, which refers to there being no difference in the cytotoxicity levels of the BJ and HaCaT cells in contact with the nonexposed (A) and exposed (B and C) CAD/CAM restorative materials powders to an acidic environment, is denied because cytotoxicity was significantly different between the tested materials on the BJ and HaCaT cell lines, thus the first null hypothesis was rejected. As regards the NO oxidative stress, the results showed no significant differences in both human BJ cells and HaCaT cells, which means that the second hypothesis is confirmed.

A potential classification regarding the biocompatibility of a dental material was defined by Atay and coworkers [50], in which the authors affirmed that cell viability above 90% means that the material is not cytotoxic. Values from 60% to 89% mean that the material is slightly cytotoxic. Values from 30% to 59% mean that the material is moderately cytotoxic. Values under 30% mean that the material is severely cytotoxic. According to the author’s classification, one can affirm that some of the CAD/CAM restorative materials tested in the present study are moderately cytotoxic (e.g., CS_C powder on BJ cells at all concentrations tested and TC_C powder, only at concentrations of 1 mg/mL and 0.5 mg/mL, Figure 6; as well as SN_B powder on HaCaT cells at a concentration of 0.2 mg/mL, Figure 7). Our results are in agreement with the literature [17,34,51,52]. An explanation of this acute cytotoxicity recorded in the case of CS restorative material, exposed to an acidic environment could be related to the ions leaching from nanoceramic resin when the CS material was immersed in a cell culture medium. In our experiment, and according to the EDX spectra (Figure 5), there are four main inorganic compounds that could be released by Cerasmart restorative material into the cell culture medium, namely silicon (Si), barium (Ba), alumina (Al) and fluorine (F). However, according to the literature, alumina and silicon are considered to have low cytotoxicity [53,54]. In addition, besides silicon, alumina is present in CS restorative material in a small weight percentage (around 2%), as well as fluorine also (under 2%). The only inorganic element which is present in a high weight percentage and is toxic to humans is barium (Ba). It is well known that Ba^2+^ ions and the compounds of them are water soluble. The only insoluble is BaCO_3_, which is soluble in the gastrointestinal tract, an acidic environment. Considering the EDX spectrum (Figure 5) of Cerasmart restorative material, one can affirm that exposure to an acidic environment led to an increased weight of Ba (CS_B powder − Wt = 17.1%). The same explanation could be in the case of TC restorative material, which presents Ba^2+^ ions in its composition and it was found that it induces necrosis at a concentration of 1 mg/mL. As regards the elemental composition of the SN restorative material tested, this has no barium in its composition, which means that the moderate toxicity of CS and TC restorative materials on BJ cells is given by Ba^2+^ ions.

Another explanation could be the impact of post-treatment after adjustment on the surface topography of a finished dental material, which includes milling, sintering, polishing, and sandblasting [55]. Due to the peaks and valleys on the material surface, left by the polishing process, oral retention could increase and thus the material surface could be damaged. It was shown that alterations in dental material composition can greatly influence cytotoxicity [56]. The above affirmation is confirmed by Dal Piva and coworkers [52], which found an increase in the initial cytotoxicity of polished dental materials, after 24 h. In addition, it was stated that the incomplete polymerization process may influence the leaching of some unreacted byproducts from the composite resin material [57,58].

The oral environment (especially the acidic one) is another factor that contributes to the degradation of the material, and thus the potentially toxic compounds leaching from the surface of the material can affect the cells [59,60]. Saliva and mastication are two fundamental factors, due to the high amount of water that contributes to the release of elements from a composite resin material [59].

HaCaT cells released less LDH after treatment with CS_C, SN_B, SN_C, and TC_B powders and even more so for the TC_C powder, at all tested concentrations. An explanation could be that the cells reacted with the milled dental materials by reducing membrane exchanges with the environment and by reducing proliferation. A slight but significant increase in the cases of CS_A, CS_B, and SN_A powders (at a concentration of 0.2 mg/mL), and for all the concentrations tested of the TC_A restorative materials. Most likely, the granules formed by milling were piercing some of the cells, determining their death by necrosis.

The reduction of LDH release for CS_C and SN_C powders to almost 0 mU/mL, correlates to reduced mitochondrial activity, indicating reduced cell proliferation. This aspect was also noticed by SEM images of the adhered cells (Figure 12). This could indicate that leaching ions and the low dimensions of the fine-granulated materials allowed cell penetration and more intimate contact with the enzymatic equipment of the cells, leading to stasis and finally cell death. TC_C powder (at concentrations of 1 mg/mL and 0.2 mg/mL) increased cells’ LDH release by 50%, which is an indication of violent cell death by necrosis. Oddly, this aspect was not observed when cells were grown on the TC restorative material slice surface, where BJ cells were better adjusted and well spread on the surface. Considering that the materials used for the cell adherence test were compact ones, without loose particles, it may be important to consider not only the ions’ leaching potential but also the material’s ability to maintain its form and structure. For BJ cells, the compact material was biocompatible and a sufficient scaffold for adherence and proliferation, however, when the same material was milled and placed in contact with them, it determined cell death.

Therefore, by corroborating our results with the results of the above-mentioned research studies, one can affirm that future studies which are evaluating the substances that are released from a CAD/CAM restorative material and causing cell damage, are important, considering that these CAD/CAM restorative materials are preferred by clinicians due to their easy preparation, polishing, and reparability features [13].

As regards the limitations of the study, one can affirm that the first limitation could refer to the cell culture testing. The experimental use cell lines, which are not identical in reactions with the tissue in site. In order to obtain more accurate results, other cell types, such as primary cells, could be helpful, or tissue culture (3D) analyses, and finally animal testing. Moreover, other analyses regarding DNA-ARN-proteins would also be useful, as well as longer time exposure analyses.

## 5. Conclusions

The current study reports the biological activity of three types of CAD/CAM restorative materials exposed and nonexposed to an acidic environment, on two normal human cell lines (BJ and HaCaT), examined at different concentrations. The outcomes reported in the present study showed that:✓As concerns the cytotoxicity evaluation, the restorative materials tested ranged from moderately cytotoxic to slightly cytotoxic to noncytotoxic on the growth of human fibroblast, according to the MTT assay and LDH assay.✓As regards the growth of human keratinocytes, the outcomes showed that the samples tested can be considered slightly cytotoxic and noncytotoxic, with the exception of SN_B, which recorded a reduced mitochondrial activity (around 50%).✓As concerns NO production, in the case of fibroblasts, this was reduced to half whereas, in the case of keratinocytes, the NO production was slightly increased compared to fibroblasts. Therefore, the tested materials do not determine cell oxidative stress on the nitric pathway.✓As regards the adhesion degree, the fibroblasts cells were more affected when the fine was exposed to an acidic environment of each material and was tested, however, when the cells were placed on each compact dental material (CS, SN, and TC), the fibroblasts cells were attached and spreading, preferring particularly the TC material.

## Figures and Tables

**Figure 1 medicina-59-00104-f001:**
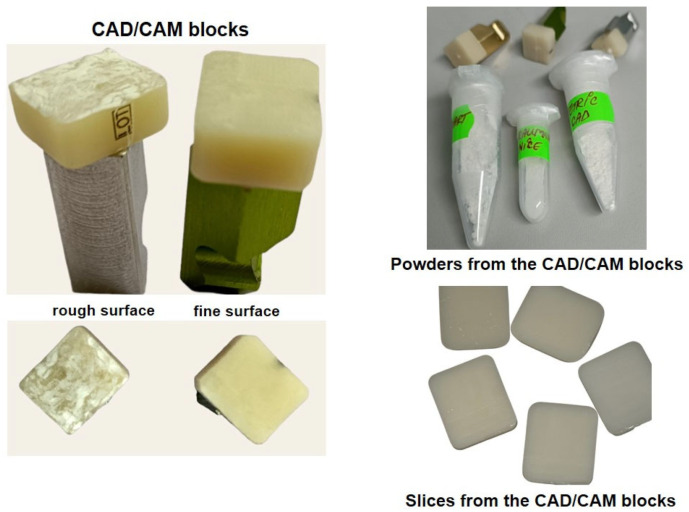
CAD/CAM blocks with rough and fine surfaces, as well as powders and slices obtained.

**Figure 2 medicina-59-00104-f002:**
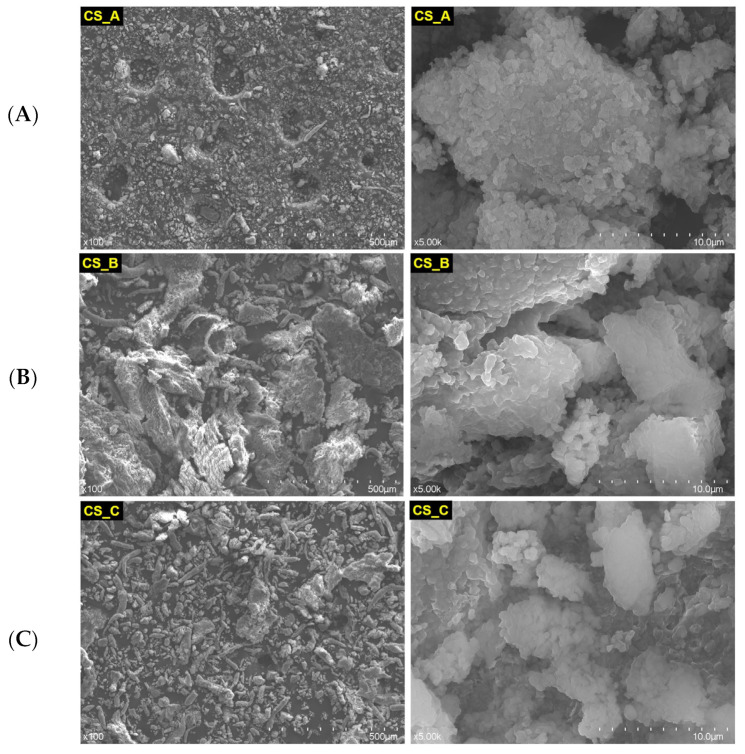
SEM images of CS restorative material powder, nonexposed (**A**) and exposed (rough (**B**) and fine (**C**)) to acidic artificial saliva.

**Figure 3 medicina-59-00104-f003:**
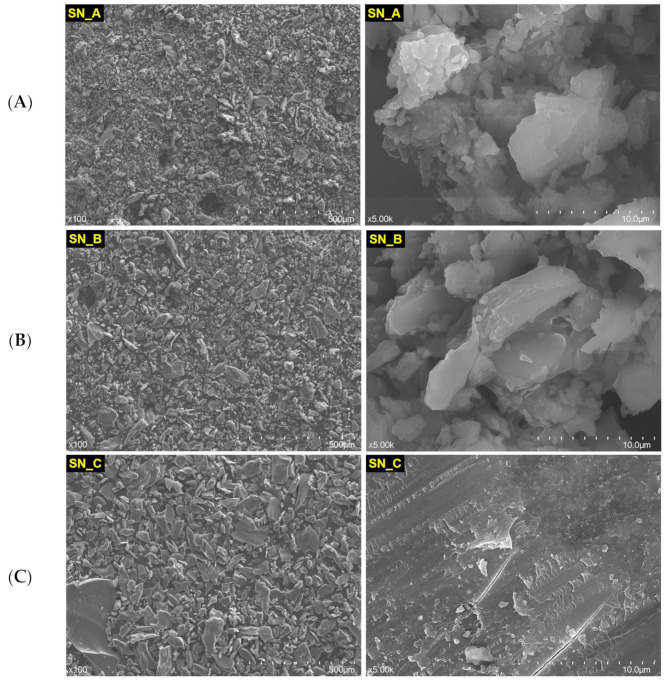
SEM images of SN restorative material powder, nonexposed (**A**) and exposed (rough (**B**) and fine (**C**)) to acidic artificial saliva.

**Figure 4 medicina-59-00104-f004:**
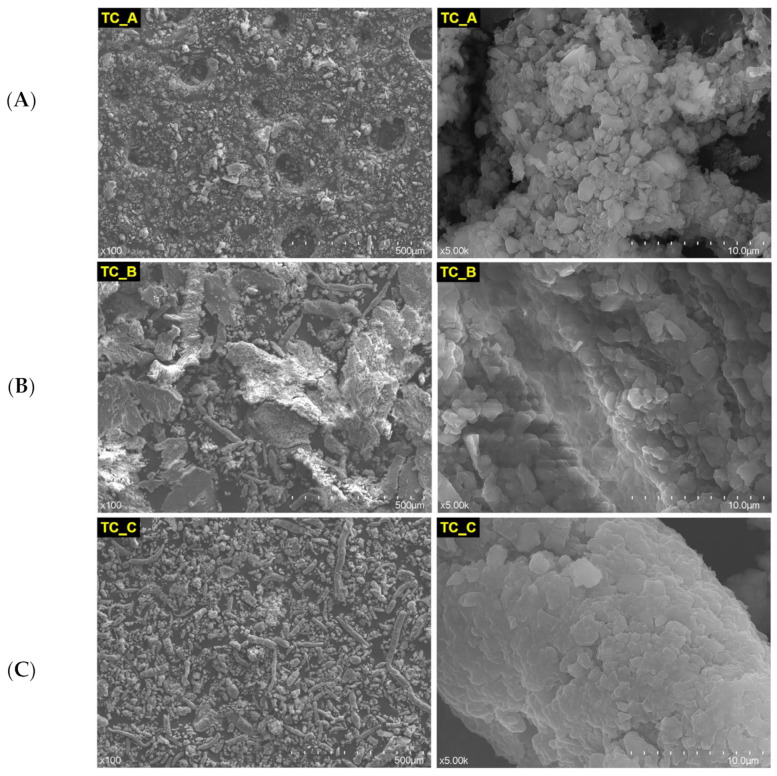
SEM images of TC restorative material powder, nonexposed (**A**) and exposed (rough (**B**) and fine (**C**)) to acidic artificial saliva.

**Figure 5 medicina-59-00104-f005:**
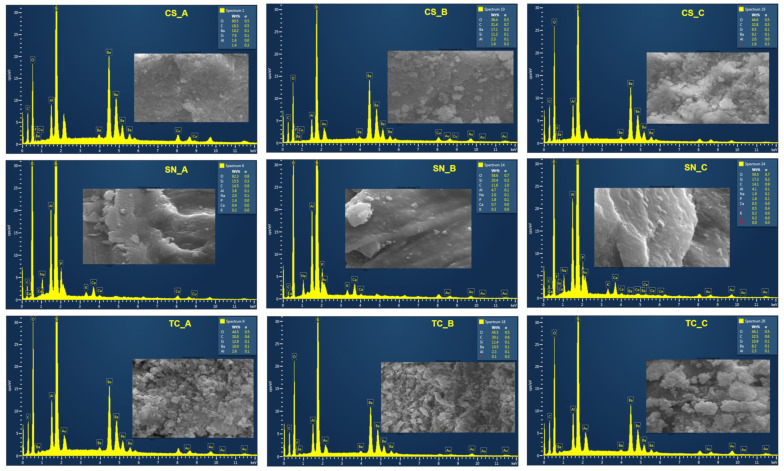
EDX spectra of the nonexposed/exposed CAD/CAM restorative materials powders to acidic artificial saliva.

**Figure 6 medicina-59-00104-f006:**
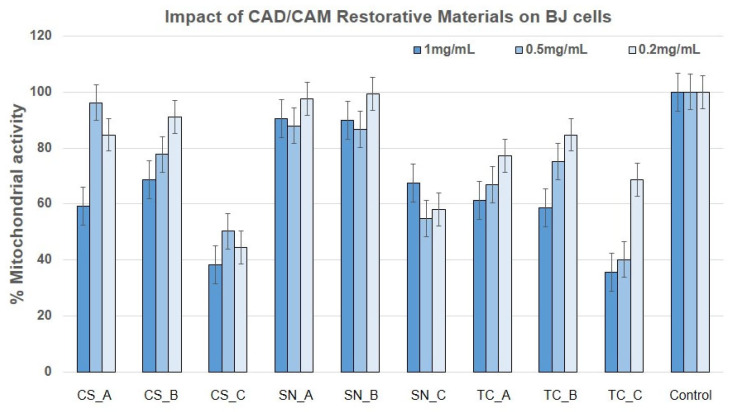
Mitochondrial activity percentage of BJ human cells, after treatment with nonexposed/exposed to acidic artificial saliva of CAD/CAM restorative materials powders.

**Figure 7 medicina-59-00104-f007:**
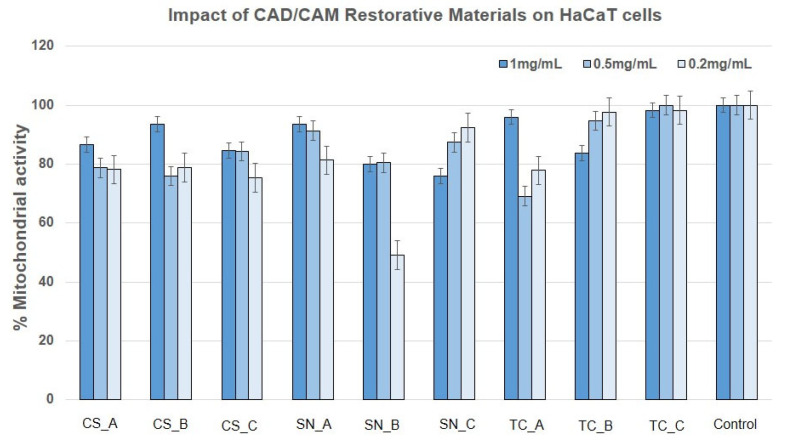
Mitochondrial activity percentage of HaCaT cells, after treatment with nonexposed/exposed to acidic artificial saliva of CAD/CAM restorative materials powders.

**Figure 8 medicina-59-00104-f008:**
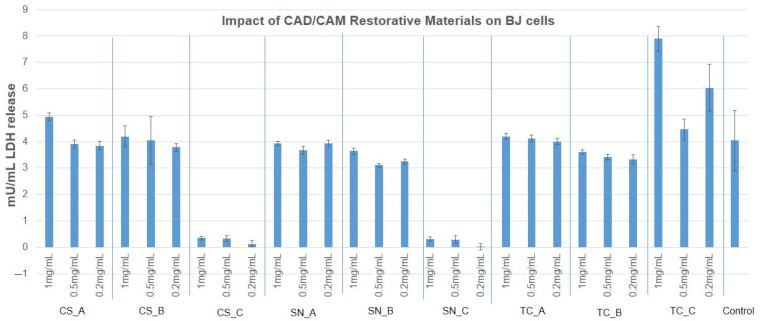
Cytotoxicity percentage of BJ cells, after treatment with nonexposed/exposed to acidic artificial saliva of CAD/CAM restorative materials powders, at three different concentrations (1 mg/mL, 0.5 mg/mL and 0.2 mg/mL), for an interval of 24 h.

**Figure 9 medicina-59-00104-f009:**
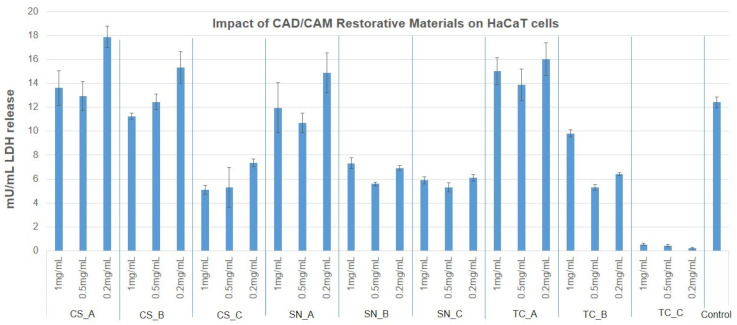
Cytotoxicity percentage of HaCaT cells, after treatment with nonexposed/exposed to acidic artificial saliva of CAD/CAM restorative materials powders, at three different concentrations (1 mg/mL, 0.5 mg/mL, and 0.2 mg/mL), for an interval of 24 h.

**Figure 10 medicina-59-00104-f010:**
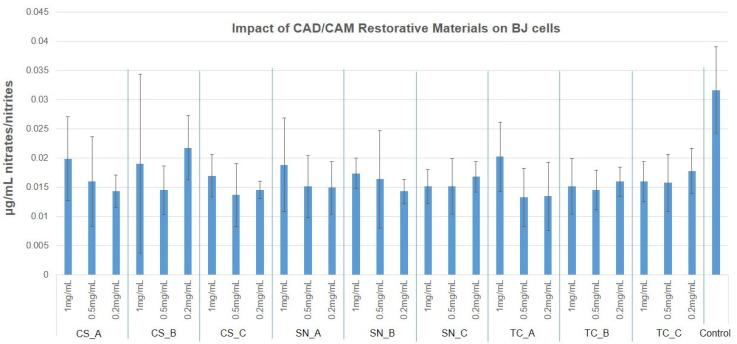
NO production of human BJ cells, after treatment with nonexposed/exposed to acidic artificial saliva of CAD/CAM restorative materials powders, at three tested concentrations (1 mg/mL; 0.5 mg/mL, and 0.2 mg/mL).

**Figure 11 medicina-59-00104-f011:**
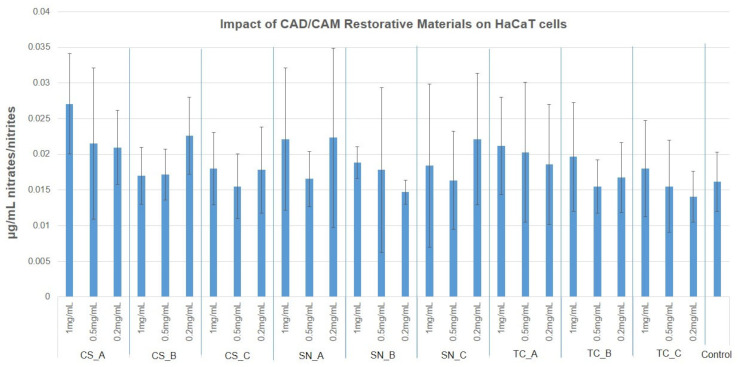
NO production of HaCaT cells, after treatment with nonexposed/exposed to acidic artificial saliva of CAD/CAM restorative materials powders, at three tested concentrations (1 mg/mL; 0.5 mg/mL, and 0.2 mg/mL).

**Figure 12 medicina-59-00104-f012:**
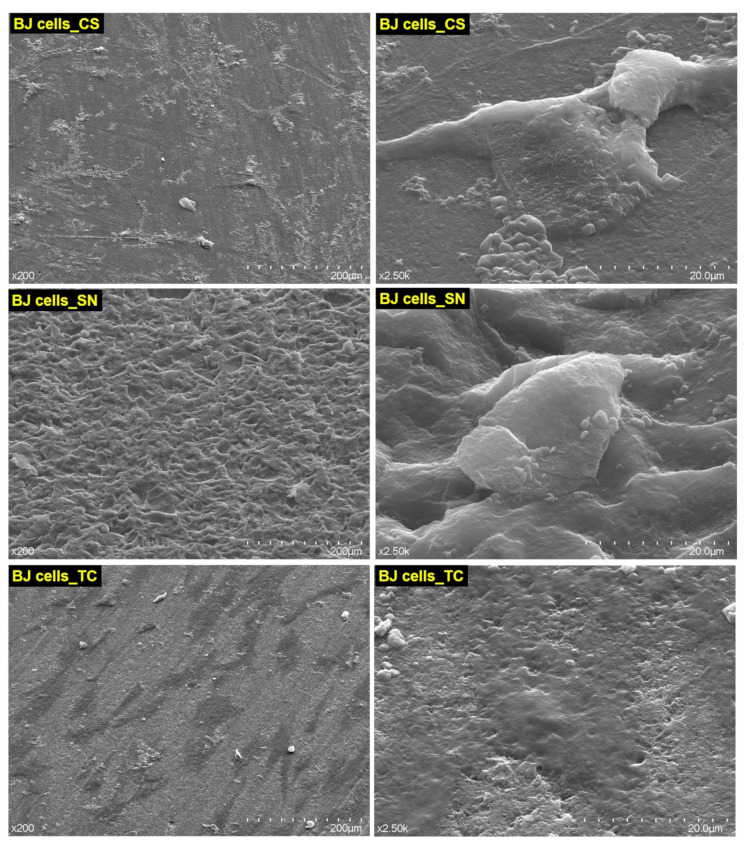
SEM images with the degree of adhesion of CS, SN, and TC–CAD/CAM restorative materials slices, to human BJ cells.

**Table 1 medicina-59-00104-t001:** Batch number, classification, and composition of the restorative materials used.

Restorative Material	Batch No.	Classification	Composition
Cerasmart (CS)	2109066	nanoceramic resin	71% silica and barium glass nanoparticles
Straumann Nice (SN)	PA354	glass ceramic	70% SiO_2_, 11% Li_2_O, 11% Al_2_O_3_, 3% K_2_O, 2% Na_2_O, 8% P_2_O_5_, 0.5% ZrO_2_, 2% CaO and 9% coloring oxides
Tetric CAD (TC)	Y07670	composite resin	barium glass (<1 mm) and silicon dioxide fillers (<20 nm)

**Table 2 medicina-59-00104-t002:** The CAD/CAM restorative materials tested in vitro.

CAD/CAM Restorative Material Group	Tested Specimen/ Denomination	Sample Description
Cerasmart (CS)	CS_nonexposed CS_A	Powder from the finished CAD/CAM block of CS, nonexposed to acidic artificial saliva
	CS_rough exposed CS_B	Powder from the rough CAD/CAM block of CS, immersed in acidic artificial saliva for 1 month, at 37 °C
	CS_fine exposed CS_C	Powder from the finished CAD/CAM block of CS, immersed in acidic artificial saliva for 1 month, at 37 °C
Straumann Nice (SN)	SN_nonexposed SN_A	Powder from the finished CAD/CAM block of SN, nonexposed to acidic artificial saliva
	SN_rough exposed SN_B	Powder from the rough CAD/CAM block of SN, immersed in acidic artificial saliva for 1 month, at 37 °C
	SN_fine exposed SN_C	Powder from the finished CAD/CAM block of SN, immersed in acidic artificial saliva for 1 month, at 37 °C
Tetric CAD (TC)	TC_nonexposed TC_A	Powder from the finished CAD/CAM block of TC, nonexposed to acidic artificial saliva
	TC_rough exposed TC_B	Powder from the rough CAD/CAM block of TC, immersed in acidic artificial saliva for 1 month, at 37 °C
	TC_fine exposed TC_C	Powder from the finished CAD/CAM block of TC, immersed in acidic artificial saliva for 1 month, at 37 °C

## Data Availability

The authors can provide raw data under request.

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
