# Peer review of "The Biological Activity of Fragmented Computer-Aided Design/Manufacturing Dental Materials before and after Exposure to Acidic Environment"

_medicina, 2023, doi:10.3390/medicina59010104_

Round 1

Reviewer 1 Report

The authors in this article provide a detailed and thorough investigation of the biocompatibility and sustainability of human fibroblasts and keratinocytes cells on ceramic and composite CAD/CAM materials. The paper provides detailed investigation very relevant to the field covered and will provide an interesting and very readable referenced text for future work. The data provided are processed in a professional manner and are well supported and integrated with the literature. Overall, the grammar and English is well written however, some areas need improvement. The authors must address some concerns with the article given below.

1.      CAD/CAM restorative materials, is the important key word to include. The authors should include it as keyword.

2.      The introduction is well written however in the literature, the author needs to focus more on CAD/CAM restorative materials, cytotoxicity and cell viability in the oral environment. Some of the relevant references are given here.

 https://doi.org/10.3390/app11198964; https://doi.org/10.1016/j.jscs.2020.01.002 ; https://doi.org/10.3390/ijms23073449 ; https://doi.org/10.1016/j.dental.2019.07.013

3.      It is suggested to resent the chemical composition of the materials and other details in a table form to make it more understandable.

4.      The method is written in very complicated way it is suggested to revise it. Specifically, the number of groups and samples in each group. It is suggested to mention it simple i.e 3n where 3 represent the number of groups and n represents the number of specimen tested for each group. The total sample size will be calculated accordingly.

5.      In the materials and method section, the first paragraph (line 145 to 151) is irrelevant. There is no need to mention the aim of the study in the materials and method section. It is suggested to remove it or shift it to the introduction.

6.      Once defined in full form, it is fine to use the CAD/CAM abbreviation in the manuscript. Same for human keratinocytes and human fibroblast.

7.      In sample preparation, it is suggested to elaborate and mention how the rough surface and fine samples were produced.

8.      For multiple samples figure please give subcategories like a, b, c etc. and mention it with information in the caption of the figures.

9.      In Figure 2 SEM images for SN-fine exposed x100, and 5.00K, there is much difference in the surface of both. It is suggested to encircle the area in low magnification image to highlight the area for image in high magnification.

10.   The authors should revise the conclusion and elaborate the main results.

Reviewer 2 Report

This is a well-designed research but the quality of the manuscript should be improved by addressing the following comments  

1. Manuscript needs professional English editing. There are errors in sentence formation etc. For example line no: 36,41,44,131….

2. Title:

Kindly modify the title, as the present study is evaluating three different CAD/CAM materials: Hybrid ceramic, Glass ceramic and composite blocks.

So please remove word “ceramic” from the title:

 The biological activity of fragmented “ceramic” computer-aided design/manufacturing dental materials before and after exposure to acidic environment.

3. Methodology:

·        Provide details about sample size and the way it was selected.

·        Please provide some pictures of the methodology

·        Provide the lot/batch numbers of the tested materials

·        The immersion regime mentioned that specimens were immersed in acidic environment for 1 month. Why this duration of 1 month was selected? Kindly provide reference for the same.  

·        Was blinding done to avoid operator bias

·        Please provide ethical committee approval details

4. Results:

As there are no limitations for page numbers in MDPI journals, Kindly provide data in tabular form too (along with graphs) so that they can be easily utilized by other authors in the future.

5. Discussion:

Please add limitations of the study

6. Conclusion:

Please provide conclusions point-wise stating each evaluated parameter.

Round 2

Reviewer 2 Report

Please provide results tables in the supplementary file.

Author Response

Tables with all the in vitro results are provided in supplementary material.